# Anti-Leishmanial Vaccines: Assumptions, Approaches, and Annulments

**DOI:** 10.3390/vaccines7040156

**Published:** 2019-10-18

**Authors:** Shubhranshu Zutshi, Sunil Kumar, Prashant Chauhan, Yashwant Bansode, Arathi Nair, Somenath Roy, Arup Sarkar, Bhaskar Saha

**Affiliations:** 1National Centre for Cell Science, Ganeshkhind, Pune 411007, India; shubhranshu@yahoo.com (S.Z.); sunilbiotech1987@gmail.com (S.K.); pra751990@gmail.com (P.C.); yashwant299@gmail.com (Y.B.); nayerarathi@gmail.com (A.N.); 2Department of Human Physiology with Community Health, Vidyasagar University, Midnapore 721102, India; roysomenath1954@yahoo.in; 3Department of Biotechnology, Trident Academy of Creative Technology, Bhubaneswar 751024, India; arup.s2010@gmail.com

**Keywords:** *Leishmania*, vaccine, antigen-specific memory T cells, antigen presentation, immune synapse

## Abstract

Leishmaniasis is a neglected protozoan parasitic disease that occurs in 88 countries but a vaccine is unavailable. Vaccination with live, killed, attenuated (physically or genetically) *Leishmania* have met with limited success, while peptide-, protein-, or DNA-based vaccines showed promise only in animal models. Here, we critically assess several technical issues in vaccination and expectation of a host-protective immune response. Several studies showed that antigen presentation during priming and triggering of the same cells in infected condition are not comparable. Altered proteolytic processing, antigen presentation, protease-susceptible sites, and intracellular expression of pathogenic proteins during *Leishmania* infection may vary dominant epitope selection, MHC-II/peptide affinity, and may deter the reactivation of desired antigen-specific T cells generated during priming. The robustness of the memory T cells and their functions remains a concern. Presentation of the antigens by *Leishmania*-infected macrophages to antigen-specific memory T cells may lead to change in the T cells’ functional phenotype or anergy or apoptosis. Although cells may be activated, the peptides generated during infection may be different and cross-reactive to the priming peptides. Such altered peptide ligands may lead to suppression of otherwise active antigen-specific T cells. We critically assess these different immunological issues that led to the non-availability of a vaccine for human use.

## 1. Introduction

The earliest vaccination protocols by Edward Jenner, Louis Pasteur, and Robert Koch demonstrated that killed, live-attenuated, or xenogenic microorganisms induced host-protective immunity [1]. During the next hundred years, the modus operandi of the immune system and the vaccine-induced protective immunity has been gradually deciphered and the central dogma of vaccination has been framed. It holds that an immunogen is capable of inducing long-lived immunological memory and protective immunity against rechallenge with the same pathogen [2]. Although this concept has been successful in generating potent vaccines for many deadly diseases, leishmaniasis caused by the protozoan parasite *Leishmania* remains a threatening exception. 

As Jose Esparaza described in the context of HIV vaccine failures, a paradigm shift from the current concept of vaccinology is essential [3]. The seven decades of failures in developing an anti-leishmanial vaccine fit this proposition very well for an urgent relook at the host–pathogen interaction dynamics [2,4]. Here, we follow the scheme of immune priming, reactivation, and outcome of challenge infection (Figure 1), emphasizing the immune response parameters that are responsible for the failures. 

## 2. Factors in Antigenic Priming That Affect Vaccination Efficacy 

### 2.1. Selection of an Infective-Stage Specific Vaccine Candidate

Leishmanization with live *Leishmania* and whole-parasite-based vaccines against *Leishmania* used promastigotes of different *Leishmania* species as a form of the vaccine but the associated risks and disadvantages prompted developing new vaccination modalities such as protein- or DNA-priming-based vaccines [5]. Due to ease in culture and characterization of promastigotes, early subunit vaccination studies primarily focused on promastigote antigens, e.g., gp63, gp46, LACK, and promastigote surface antigen-2 (PSA-2) [6,7]. As *Leishmania* amastigotes propagate in humans [8] and as its proteome is available for degradation and presentation by antigen-presenting cells (APCs), an anti-amastigote immune response is vital for the maintenance of long-term immunity. Immunomodulatory Th2 properties of sand fly saliva [9,10,11] prompted the inclusion of genes for sand fly salivary proteins in the vaccine construct. Polarization of Th subsets to Th1 and Th2 in *Leishmania*-resistant C57BL/6 and *Leishmania*-susceptible BALB/c mice, respectively, indicates that the *Leishmania* antigen-specific Th1-clones at later stages of infection, via amastigote degradation, may significantly contribute to protection and disease progression. Many amastigote-specific antigens including A2 protein from *Leishmania donovani* [12], hydrophilic acylated surface protein B1 (HASPB1) of *L. major* [13] and *L. donovani* [14], p27-/-, and LdCen-/-*Leishmania* [15,16] were discovered as probable vaccine candidates. Table 1 presents a comprehensive compilation of the vaccine approaches and analyses.

### 2.2. Determinant Selection as a Function of Relative Abundance of Leishmanial Proteins

Availability of an intracellular protein for protease-dependent degradation depends on its abundance and expression inside the APCs. Analysis of 2.5% of the most abundantly expressed mRNAs showed that nearly 41% and 11% of encoded proteins from these mRNA’s contained MHC-I and MHC-II ligands, respectively [115]. High-throughput MS-based analyses of 22,000 HLA-binding peptides across cancer cell lines and primary cells identified a significant positive correlation between a protein’s expression level and its presentation though HLA, where a faster degradation of overexpressed proteins was found to be in part responsible for their preferred presentation to MHC molecules [116]. Therefore, the antigen-specific recall response generated during the infection would depend upon the intracellular expression level of the immunogenic stage-specific protein used for priming, as it may be outnumbered by increased turnover of other proteins in infected macrophages affecting their loading onto MHC-II molecules. Intracellular competition between the potential determinants would thus play a key role in the expansion of the antigen-primed host-protective memory T cells during infection.

### 2.3. Intracellular Compartmentalization Contributes to the Processing of an Antigen 

Even if *Leishmania*-infected macrophages form phagolysosomes, the availability of antigen for protease-dependent processing would depend on its subcellular localization in *Leishmania. Leishmania* cysteine protease-specific Th cell lines were better activated by macrophages containing inactivated or killed parasites [117]. Macrophages containing live parasites ectopically expressing Leishmanial membrane-bound acid phosphatases (MAPs) either on parasite’s surface or insoluble form were able to better activate T cells; however, wild-type MAP and cysteine proteases expressing *Leishmania*-infected macrophages were unable to activate T cells [118]. Amastigotes are able to sequester and protect the protein antigens from uptake and degradation by MHC-II presentation pathway, possibly by modification of cellular trafficking pathways [119]. Splenic DCs efficiently presented and elicited immune responses to surface localized OVA rather than cytoplasmic form when infected with OVA-expressing transgenic *Leishmania* [120]. Antigen localization plays a crucial role in its uptake and presentation via MHC-II in the altered physiological environment of *Leishmania*-infected macrophages for the induction of immune responses and reactivation of the antigen-primed memory T cells during exposure to live virulent *Leishmania*. 

## 3. *Leishmania* Targeting of Antigen Processing and Presentation Affects Vaccination Efficacy

### 3.1. Receptor-Mediated Internalization of Leishmania Parasites

Receptor-mediated endocytosis of *Leishmania* by macrophages involves numerous receptors, e.g., complement receptor 1 (CR1), CR3, Fc-gamma receptors (FCγR), and fibronectin receptors (FnRs), which assist interaction or docking of parasites on macrophage surface [121,122]. The complement components are endogenous adjuvants for vaccine-induced CD8+ T cell responses in *Leishmania* infection [123]. Due to their phenotypic plasticity, M1 and M2 macrophages change phenotypes during *Leishmania* infection [124]. As lipophosphoglycan (LPG)-dependent TLR2 activation during *Leishmania* infection causes M1/M2 polarization of macrophages altering cytokines stoichiometry, macrophage plasticity is attributed to cytokines; particularly, IFN-γ promoted M1 and IL-4, IL-13 driven M2 phenotypes (Figure 2). 

M1 macrophages are potent producers of reactive oxygen species (ROS), inducible nitric oxide synthase (iNOS), and reactive nitro-species, and also function as effective APCs, secreting high levels of IL-12 and IL-23 [125]. These observations imply that Fc glycosylation and FcR/CR interactions during priming may be manipulated to establish robust macrophage polarization. 

### 3.2. The Hijacking of Lysosomal Fusion Machinery inside Macrophages

Phagolysosome biogenesis is the major defense mechanism of immune cells from invading pathogens. Synaptotagmin (Syt) Type-1 membrane protein regulates vesicular fusion processes such as exocytosis and phagocytosis. While SytV is a major regulator of phagolysosome biogenesis, SytXI is involved in secretion from *Leishmania*-infected macrophages in gp63-dependent manner [126]. SytV silencing inhibits the recruitment of cathepsin D and vesicular proton ATPase to endosomal membrane. LPG binds to phagosome membrane microdomains and dissociates SytV hampering the acidification process, phagolysosome formation, and aiding parasite survival [127]. A host-protective role of transcriptional regulator STAT1 in maintaining counter ion conductivity by chloride ion channels, which are important for phagosomal acidification, and macrophage proteasomal system, are reported [128,129]. Hence, phagosomal maturation is a step, which *Leishmania* targets initially to survive inside macrophages until the time it transforms into acid-resistant amastigote form, inhibiting the antigen presentation process and affecting the vaccine-primed antigen-specific T cells’ reactivation (Figure 3).

### 3.3. Altered Activity of Proteases

*Leishmania* infection tweaks the proteasomal machinery to increase the degradation of transcription factor STAT-1α and reduces the release of interferons [128]. Lysosomal proteases like cathepsin generate high-affinity peptides for efficient binding to MHC [130]. Due to their selectivity in cleavage sites, selective inhibition of these proteases during infection can change the type of peptide generated in APCs. While IFN-γ stimulation increased cathepsin D activity, LPS stimulation had the opposite effect on cathepsin D [131]. As LPS works through TLR4, TLR4 activation by *Leishmania* [132] may alter cathepsin S activity by LPS and IFN-γ mediated alteration in cathepsin S expression [133]. The repertoire of T cells generated during priming and natural infection by the same antigen may thus change due to parasite-induced selective interference in proteolytic processing of antigens. 

### 3.4. Degradation of Antigen Presentation Machinery

During early MHC biosynthesis, I chain is attached to MHC-II αβ dimer to prevent premature peptide loading. Proteolysis and acidic pH induce I chain release and replacement with CLIP (class II-associated invariant chain peptide). When MHC-II is ready for antigen loading in mature or late endosomes, CLIP is dissociated from MHC-II via another glycoprotein DM that induces a conformational change in CLIP molecule [134]. Both MHC-II and H2-DM molecules are taken up by amastigotes within parasitophorous vacuoles resulting in degradation of MHC-II and associated molecules [135], and hindering the antigen presentation to T helper cells affecting their activation and differentiation into cytokine-secreting effectors. 

### 3.5. Immunodominance and Epitope Crypticity: Importance in Anti-Leishmanial Vaccination

In spite of the above-mentioned problems with antigen processing, the selection of the priming epitope may face altered immunodominance during *Leishmania* infection. Immunodominance refers to the “preferred” or “biased” selection of a few epitopes of a protein or from a whole proteome for MHC-I/II presentation, during the generation of an anti-pathogenic immune response. An immunodominant epitope elicits a higher response when compared to a subdominant epitope. The epitopes that are not “favored” for binding to MHC molecules are termed as cryptic epitopes. There are many factors that can influence the dominance or crypticity of a particular T cell epitope: (1) TCR recognition sites or intrinsic nature of antigen, (2) peptide-MHC-II affinity, (3) abundance of particular antigen, (4) promiscuous binding of antigen, and (5) MHC-guided processing and the flanking regions of peptides [136,137]. Vaccination with LACK antigen from *L. major* causes a CD8+ T cell-dependent protection against challenge with *L. major* [78,138,139]. The expression of Pgp-1 glycoprotein enriched antigen-stimulated memory T cells within the periphery [140] but their sustenance requires persistent antigenic stimulation [141]. Thus, the antigenic crypticity or shifting immunodominance during antigen presentation in *Leishmania* infection plays important roles in vaccine outcome.

### 3.6. Loading of Antigen to MHC Molecule 

*L. donovani*-infected THP-1 macrophages have a different HLA-I self-peptide repertoire [142] implying that *Leishmania* infection may affect the antigen-specific T cell repertoire. The incapability of *Leishmania*-infected macrophages to activate Th cell was reversed by low pH or 2-(1-adamantyl)-ethanol treatment altering intracellular conditions favoring peptide exchange and a significant increase in anti-Leishmanial T cell compartment, suggesting binding of incognate peptides hindered the loading of Leishmanial peptides during the course of infection [143]. Hence, a perturbed MHC compartment inside the *Leishmania*-infected macrophages can lead to the inefficient presentation of anti-leishmanial peptides (Figure 3).

### 3.7. Peptide-MHC-II Affinity: A Major Determinant of Immunodominant Epitope Selection

The most significant factor for the competent presentation of a foreign peptide to a Naïve T cell and immunodominance of the selected peptide [144] is a peptide’s affinity for the peptide-binding groove of an MHC molecule. Importance of a single residue in the binding potential of an antigenic peptide was described where substitutions in *S. aureus* nuclease epitopes can convert immunodominant epitopes into subdominant ones and vice-versa, and this principle is applied to cancer therapy using *Pseudomonas* exotoxin-A [145,146,147]. These findings have significant implications in vaccine development due to following reasons: (1) Change in amino acid residues of flanking regions of epitopes may render them less susceptible to action of proteases and alter their processing for efficient presentation by APC; (2) substitution in anchor residues of a peptide can decrease the affinity of an otherwise immunodominant epitope and it can become subdominant or cryptic; (3) a set of mutations in the protein sequence may modify its natural conformation, thereby rendering the protease-sensitive regions inaccessible to the enzymatic breakdown (Figure 3). These events change the repertoire of epitopes being presented on APC for T cell activation impairing the recall response to a vaccine candidate during natural infection.

Recognition of specific cleavage sites on epitopes by proteases is an essential step during antigen processing. Schneider et al. increased the presentation efficiency of a subdominant epitope of hen egg lysozyme by adding a dibasic endopeptidase motif in its flanking regions; however, the T cell stimulatory capacity remained unchanged [148]. Substitutions altering processing and presentation of epitopes from *Pseudomonas* endotoxin-BII domain reduce their immunogenicity [149]. Spacer sequences or various tags added to recombinant proteins during purification steps were shown to distort the natural dominance of specific epitopes [150]. A change at a single site on a peptide can drastically change its presentation by APC. This is crucial in the context of *Leishmania* as there are copy number variations (CNV) in various genes of protozoan parasite associated with drug resistance [151,152]. This process can obviously generate proteins, which may be functionally similar, albeit possessing different amino acid residues generating from duplicated gene loci. This, in turn, may affect the antigen processing dynamics of a peptide and change the repertoire of physiological immunodominant epitopes of a given antigenic protein.

## 4. Leishmania Targets Antigen Presentation by APCs and Immune Synapse (IS)

The dynamics of APC–T cell interaction, expression levels of co-stimulatory and co-inhibitory molecules, and the cytokine milieu play decisive roles in the anti-leishmanial immune response. In the lymph node, T cells move through a meshwork of APCs, and on recognizing MHC-bound cognate antigen, forms a contact zone—immune synapse (IS)—at the T cell/APC interface. Exposure to chemokines during extravasation activates the adhesion molecules on T cells, leading to the interdigitation of cell’s glycocalyx. Adhesion occurs mainly through T cells expressed α_L_β_2_ integrin LFA-1, which upon activation interacts with ICAM-1 expressed on endothelial cells. TCR signaling micro-clusters forms at the periphery of IS, which gradually moves inward by the retrograde flow of actin cytoskeleton and coalesce to form the main contact zone or core supramolecular activation cluster (cSMAC). cSMAC comprises of TCR-MHC complex enriched with Src-family kinases like Lck and Fyn, CD28, protein kinase C θ (PKCθ) and zeta-chain-associated protein kinase 70 (ZAP-70). In addition to this, interaction occurs at segregated contact zones or peripheral SMAC (p-SMAC) via LFA-1/ICAM-1, CD2/CD58 interaction [153,154]. pSMAC also has the cytoskeletal linker protein talin. The outermost distal SMAC (dSMAC) has CD43 and CD45 and performs mechano-supportive role. These molecular interactive forces are at peak after 30 min [155]. Once the T cell is in contact with the APC, TCR conveys a stop signal, forming a stable junction IS [156]. As an effect of LFA-1 and ICAM-1 interaction, downstream signaling events result in cytoskeletal remodeling and T cell motility [157] (Figure 4).

On TCR stimulation, several signaling proteins like the adaptor protein Linker of activation of T cells (LAT), which is phosphorylated by ZAP-70 [158], PKCθ [159], and tyrosine kinase Lck [160] are targeted to cholesterol and sphingolipids rich lipid rafts. *L. major* depletes membrane cholesterol [161] that may affect the signaling component’s segregation in lipid raft and the formation of a stable IS. *L. donovani* also interferes with the engagement of MHC-peptide complex with TCR [162], possibly via alteration of membrane lipid content, as shown by protection against VL by hypercholesterolemia [163], by cholesterol-enabled formation of a stable IS and cross-presentation and activation of CD8+ T cells [164]. *Leishmania* alters membrane cholesterol, disrupting the stability of the MHC-peptide complex, MHCII conformation, and affecting antigen presentation [165,166]. A cholesterol-liposome treatment re-established the formation of immune synapse, restored anti-leishmanial T cell repertoire, and immune functions in *L. donovani* infection [167]. 

Expression of cell surface molecules and their interaction also dictates the robustness of T cell activation. *Leishmania donovani* decreases the expression of both MHC-I and MHC-II in macrophages [168] and DCs [169] by direct internalization [135] or by the action of certain extracellular vesicles [170]. *Leishmania* binds to and affects the functioning of integrins [171] and αMβ2 (Mac-1) [172] on phagocytes, impairing IS formation and altered T cells stimulation. *L. amazonensis* also impairs the formation of adhesion signaling complexes on the plasma membrane [173]. 

The expression of co-stimulatory molecules and the threshold and magnitude of their interaction with co-receptors on T cells [174] can influence the generation of antigen-specific T cells. *L. donovani* lowers the expression of co-stimulatory molecules CD80 and CD86, which are critical for Th2 differentiation [175,176]. It also alters the micro-domain lipid constitution, hence lowering the availability of TCR ligands on the surface of macrophages [162]. CD40 and CD80 play a crucial role in anti-*Leishmania* T-cell activation, wherein *L. major*-infected macrophages co-cultured with peripheral blood leukocytes (PBL) showed an IFN–γ- and CD40-mediated increase in the expression of CD40 and CD80 [177,178,179,180,181]. 

CD4+ and CD8+ T cells express inhibitory cytotoxic T-lymphocyte-associated protein 4 (CTLA-4/CD152), which binds to CD80 (B7-1) and CD86 (B7-2). It terminates T cell response, generation of memory T cells [182], and regulates T cell anergy [183]. The maximum level of expression of CTLA-4 occurs 24–72 h post-activation [184]. Blockade of B7-2 following *L. donovani* infection resulted in enhanced Th1 and Th2 responses [185]. CTLA-4 blockade increased IFN-γ, IL-4, and CXC chemokine γIP-10 [186]. PD1 (CD279), an inhibitory co-receptor, is recruited to the IS and its engagement with PDL1 or PDL2 leads to the recruitment of tyrosine phosphatase SHP-2, that attenuates TCR signaling by phosphorylating signaling molecules like Vav, Akt, CD3ζ, ZAP70, and ERK [187]. *L. donovani*-infected macrophages showed a reduced PD1 expression and inhibition of apoptosis [188]. Another immune-inhibitory receptor on macrophage is the signal regulatory protein (SIRP)-α [189] but its role in T cell priming remains unknown. 

Transcriptional profiling of genes upregulated during *L. infantum* infection shows that there is a marked difference in the gene expression in infected mice when compared to control mice. Upregulation of Th1 and M1 macrophage activation markers like *Ifng, Stat1, Cxcl9, Cxcl10, Ccr5, Cxcr3, Xcl1*, and *Ccl3* occurs in early infection. Of the 112 genes studied, 22 showed differential expression in infected vs. uninfected mice. *Il12rb2*, *Il23r*, and *Ptgs2* were exclusively expressed only in infected mice but not in the control group, suggesting that the immune molecules may vary drastically under the condition when initial priming is done (uninfected) compared to when subsequent infection occurs [190]. Thus, an aim in vaccination is to induce the balanced expression of co-stimulatory and co-inhibitory molecules such that appropriate Th cell priming occurs.

As the IS forms, several cytokines are secreted at the IS. The Th cells secrete cytokines such as IFN-γ, IL-10, and IL-2 [191] towards the antigenic target cells, or cytokines like IL-4, TNF, CCL3, MIP-1α towards target as well as non-target cells [192]. Actin remodeling plays an important role in the secretion of cytokines at the IS as silencing of Cdc42, a regulator of cytoskeletal rearrangement, inhibits the secretion of cytokines like IFN-γ at the IS [193]. 

During APC–T cell interaction lasting about ~12 h, abundant IL-2 is produced [194]. However, many other cytokines secreted over time, which may not be present at the time of IS formation, may be important in limiting *Leishmania* growth. Due to limited literature on this topic, one might speculate the potential cytokines trafficking from T-cells in case of *Leishmania* infected macrophages and vice versa that shapes the immune synapse are yet to be uncovered and many can have profound effects on both cell types and antigen presentation at IS. Since IS formation is a regulated, dynamic, and sequential process [195], it has multiple points of disruption that intracellularly residing parasites from *Trypanosomatidae* family may exploit to ensure their intracellular survival. MHC-II peptide density plays an important role in driving signals through TCR and subsequent cytokine secretion. The cytokine expression of a particular T cell subset may be determined by the affinity of its TCR for peptide-MHC-ligand [196]. Therefore, the binding strengths may be one of the mechanisms through which the cytokines may get affected at IS. For example, the coinhibitory receptor CTLA-4 has a higher affinity to bind to CD86 [197] than CD28, therefore, it outcompetes CD28 binding at IS and may lead to a suppressed T cell instead of activation [198]. In the case of murine VL, CTLA-4 increases TGF-β levels and apoptosis of CD4+ T cells [199]. The administration of anti-CTLA-4 mAbs to BALB/c mice following one day of infection with *L. donovani* enhances the frequency of IFN-γ- and IL-4-producing cells in both spleen and liver contributing to resistance against the parasite [186]. Aided with the availability of recombinant cytokines along with the monoclonal Abs, targeting inhibitory receptors provide some promising tools with which the limitations surmounting anti-leishmanial vaccines can be overcome. 

## 5. Various Fates of T Cells during Infection

T cell response to an infection is a dynamic process. Following vaccination, phenotypic changes occur in antigen-specific T cells, which may be further altered during the course of infection [200]. The ‘functional avidity’ determined by the strength of ligand–receptor interactions at the IS or the T cells responsiveness to the titer of cognate antigen in vitro or their activation threshold is another parameter crucial in designing effective vaccines [201,202,203]. Polymorphisms in the co-receptor [204], presence of inhibitory molecules [205], micro RNAs [206], and location of the ligand [207] can influence T cell responses. On impaired antigen presentation and an unstable IS formation, only a few T cells are activated. These T cells can have different fates and can undergo anergy, exhaustion, apoptosis, or can induce a non-robust T cell response due to T cell plasticity. 

### Plasticity of T Cells 

Naïve CD4+ T cells can be induced to differentiate into Th1, Th2, Th9, Th17, Th22, T follicular helper (Tfh), and T regulatory (Treg) cell type, based on the cytokine milieu. IL-12 signals through STAT4 and induces differentiation into Th1, and IL-4 signals through STAT-6 and skews the differentiation into Th2. TGF-β and IL-6 activate Smad family proteins and STAT3, respectively, driving Th17 differentiation, whereas TGF-β and IL-4 towards Th9, IL-6, TNF- α towards IL-22, IL-21 towards Tfh, and TGF-β alone towards Treg. These cells play decisive roles in modulating the efficacy of vaccines. Th17 are inflammatory cells that produce IL-17A and offer protection against intracellular pathogens, whereas Tregs play an immunosuppressive role and dampens the immune response induced by vaccines. TGF-β skews the T cell response to either Th17 or Treg [208]. The high concentration of TGF-β is conducive for Treg, whereas the low concentration of TGF-β, along with cytokines such as IL-21 and IL-6, promotes Th17 responses [209]. Treg cells express high-affinity IL-2 receptors and IL-2 stabilized Foxp3 a transcription factor crucial for maintaining suppression of the immune system [210]. However, the presence of proinflammatory cytokines like IL-21, IL-6, IL-23, and IL-1β, promotes differentiation into Th17 rather than Tregs [211]. Treg-induced immunosuppression is effected by both cytokines and co-inhibitory molecules like IL-10, TGF-β, perforins, granzymes, and CTLA-4 and LAG3 [210]. The Th17 differentiation to induced Treg (iTreg) can be also reprogrammed by small-molecule-like (aminooxy) acetate [212]. The plasticity of T cells, deviating from one subset to another based on the cytokines present during infection, thus induces a non-robust T-cell response.

## 6. Anergy, Exhaustion, and Apoptosis

Even in case of a robust T cell response, complete parasite elimination may not happen, resulting in chronic dormant infection. Under such an infection, exhaustion or attenuation of effector T cells occurs [213]. T cell exhaustion is characterized by low proliferation, production of IL-2, and inability to produce TNF-α and IFN-γ. A high load of antigen during the course of infection and high-affinity interaction can lead to rapid CD8+ T cells proliferation whereas lack of integration of stimulatory signals can drive CD8+ T cells to exhaustion [214,215], anergy, or apoptosis [216]. Of these integrating stimulatory signals, the cytokine milieu is a major determinant of the proliferation rate of naïve and memory T cells [217]. T cell anergy can occur due to the APC expressed immunomodulatory tryptophan metabolizing enzyme indoleamine-2,3-dioxygenase (IDO) [218] or ATP-catabolizing enzymes CD39 and CD73 [219]. Anergy can also occur when T cells interact with an immature APC, whereby the TCR–peptide–MHC interaction takes place but adequate costimulatory signals are not obtained [220]. Treg cells expressing immuno-modulatory molecules like IDO [221], ATP-metabolizing enzymes [222], and CTLA-4 [223] can result in sustained T cell anergy. The increased expression of IDO [224] or ATP-metabolizing enzymes [225] on DCs during *L. major* and *L. amazonensis* infection leads to poor induction of T cells. In *L. donovani* infection, an increased expression of PD-1 on parasite-specific CD8+ T cells characterizes T cell exhaustion [226]. *L. amazonensis*-infected macrophages produced TGF-β, a pro-apoptotic, immunosuppressive cytokine that induces T-cell anergy/apoptosis [227]. 

Parasite-specific T cells that surpass all stages of infection undergo programmed cell death once the infection is cleared, with an exception of a few memory T cells that have an antigen-independent long-term survival [228]. Further increase in the number of antigen-specific CD8+ T cells can be achieved via booster immunization. The nature of antigen used for booster immunization is of prime importance, as boosting with the homologous antigen can lead to its neutralization and dampened CD8+ T cell responses. This can be overcome by the use of a heterologous antigen [229]. However, repeated boosting can lead to the terminal differentiation and unresponsiveness of the memory T cells [230,231]. Repeated immunization has shown to increase the functional diversity of memory T cells and decrease their proliferation rate [232]. The quality of memory T cells generated under such conditions depends on the time duration between the immunization, the inflammatory status, the characteristics of the antigen(s) used for immunization, and the responses of the immune system towards each antigen [233]. As a consequence of re-stimulation of T cells, activation-induced cell death (AICD) or T cell-autonomous death (ACAD) may occur via expression of ligands like FasL, CD95L, TNF, etc. or by increased expression of Bim in T cells [228] leading to apoptosome formation [234]. As the inflammation is high during priming, it leads to the accumulation of effector and memory T cells and antagonistically delays the conversion of early CD8+ cells to late memory T cells [235]. 

### Leishmania-Infected Macrophages Respond Differently to Antigen-Specific T Cells

Activated T cells capable of killing *Leishmania*-infected macrophages exit the lymph node and reach the site of infection and secrete cytokines like IFN-γ. Many studies discuss that the refractoriness towards IFN-γ abundance may be attributed to IFN-γR inhibition, as IFN-γ leads to classical (M1) macrophage activation generating nitric oxide and other microbicidal free radicals responsible for *Leishmania* killing. *L. donovani* amastigotes are capable of drastically reducing the levels of IFN-γ-induced expression of MHC-II and iNOS expression. This causes a fragmented anti-parasitic response via-JAK-STAT signaling. *L. donovani* amastigotes are also reported to impair IFN-induced STAT1α nuclear translocation by abrogating its association with importin-α5 [236]. *L. major* and *L. mexicana* modulate IFN-signaling, by suppression of IFN-γRα and IFN-γRβ subunit expression in addition to the components of this pathway such as Jak-1 and Jak-2 [237]. Apparently, this effect was attributed to the decreased availability of the receptor protein after infection [238]. SHP-1 phosphatase is another regulator of cytokine response in infected macrophages, as deficiency of this phosphatase leads to increased inflammatory cytokine response (TNF-α, IL-1, and IL-6). *Leishmania* can control SHP-1 activity and can allow its intracellular survival. *Leishmania*-induced increases in activation of macrophage SHP-1 are associated with impaired IFN-γ-triggered JAK2 activation [239]. In such a condition during the actual course of *Leishmania* infection, the infected macrophages would be unresponsive to the available pools of IFN-γ or TNF-α. Because a small number of antigen-specific T cells overcome the first barrier of antigen selection in situ, the host-protective anti-leishmanial effects are not abundant. Thus, abrogation of IFN-signaling, macrophage activation, and Jak/STAT signaling [240] affect the vaccine performance and possible accentuation of this pathway may bring therapeutic benefits to anti-leishmanial vaccination programs. A representation of selective-modulation of IS and its components by *Leishmania* is indicated in Figure 4. 

## 7. Conclusions

Since the demonstration of vaccinating principles by Jenner, researchers have followed the same for protecting vaccines. Although vaccinating with whole organisms—killed or attenuated in many different forms—brought success in many cases, the anti-leishmanial vaccines failed. Similar can be said of recombinant or peptide vaccines. As all the approaches with assumptions that introduction of the antigen(s) to a host will elicit host-protective immunity against the virulent pathogen turned out to be invalid, we took up this analytical review to find out the possible reasons for the failures and the possible approach that may bring success to *Leishmania* vaccine generation. Firstly, trafficking of T cells between lymph nodes and site of infection is essential for their activation and maturation. Inhibition of CD4^+^ T cell migration pattern in lymph nodes is reported during *Shigella* infection, which affects antigenic priming. Similar reports are found in HIV-1 infection, whereby continuous stimulation of immune system impairs the migration ability of circulatory CCR6+ CXCR3+ Th cells from blood to peripheral organs [241,242]. In fact, *Leishmania* infection impairs the migration of DC’s from the site of infection to lymph node for antigenic priming, which alters host immune response [243,244]. Therefore, thorough assessment of immune priming, circulating and central memory T cells are crucial for vaccination-induced disease protection. Secondly, epigenetic regulation of host-protective signaling in *Leishmania*-infected macrophages [245] indicates that parasite-induced epigenetic events may also be responsible for Th cell plasticity, altered MHC presentation, and vaccination outcome. Thirdly, there lies ambiguity in the importance of persistent parasites in maintaining central memory T cells. It needs to reassessed from a new perspective as it may lead us to revisit whole-parasite-based vaccines as a potential vaccination strategy [246,247]. In fact, killed-parasite vaccine was found to be inefficacious due to early recruitment of neutrophils, and healed C57BL/6 mice were immune to subsequent sand fly challenge infection [248]. This fortifies the role of additional parameters, such as sand fly salivary proteins in conferring protective immunity as shown in other studies [249]. Also, adoption of sand fly challenge model is the need of the hour for a better correlation of immune response parameters. Glennie et al. showed that T_RM_ (tissue-resident memory) cells reside in tissues distant to *Leishmania* infection site and are better equipped at controlling subsequent infection by quick recruitment of circulating memory T cells in CXCR-3 dependent manner. However, later observations indicated that protection by T_RM_ cells in *Leishmania*-immune C57BL/6 mice is mediated by early recruitment of inflammatory monocytes, which eliminate *Leishmania* via ROS and NO generation. Irrespective of effector mechanism, T_RM_ cells, a novel memory T cell subset, appears to be important in generating protective immunity during the early phase of infection, when parasite immune-escape mechanism are relatively dormant [250]. This needs detailed experimental analysis as the ability to generate T_RM_ by vaccine candidates may be incorporated as a novel parameter for assessing vaccination efficacy. Fourthly, re-enforcing parameters such as dendritic-cell based vaccine, TLR ligands-based adjuvants, and supplementing the vaccination with a blockade of co-inhibitory signal molecules like CTLA-4, PD-1, CD200R, and TIM-3 deserve consideration for inclusion in the vaccination protocol. It may enhance vaccination efficacy and T cell activation by modulating B7 molecules [198]. Finally, the principle of vaccination that focuses on host-protection as the outcome needs to include the principles of immunoregulation so that the elicitation of priming-induced T cell memory should be robust enough to match the immune interfering effects of the exposure to a pathogen. Hence, reassessment of anti-leishmanial immune-correlates in vector-transmission based infection models, incorporating T_RM_ cells in pre-clinical vaccine efficacy studies, and co-inhibitory signal blocking at IS during prophylaxis look to be promising future approaches for reaching the goal of a successful anti-leishmanial vaccination.

## Figures and Tables

**Figure 1 vaccines-07-00156-f001:**
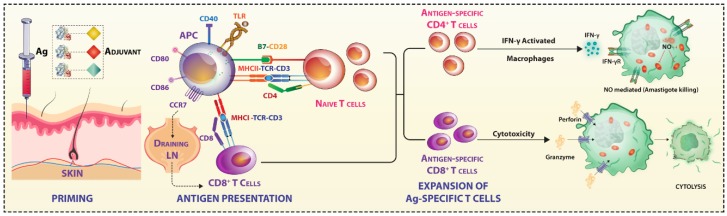
*Central Dogma of Vaccination* maintains that the host-protective T cells elicited by optimal immunization protocol protect the host from developing disease upon exposure to the pathogen. However, in the case of Leishmania, all the protocols have failed so far in protecting human vaccines.

**Figure 2 vaccines-07-00156-f002:**
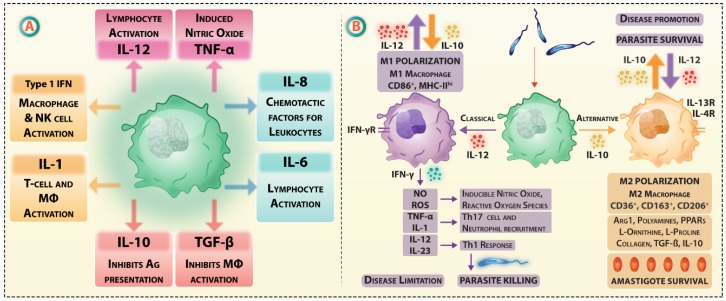
**(A**) Cytokines secreted by macrophages and their effects on immune system; (**B**) M1 and M2 type macrophages polarization in Leishmania infection and its implication on disease pathogenesis.

**Figure 3 vaccines-07-00156-f003:**
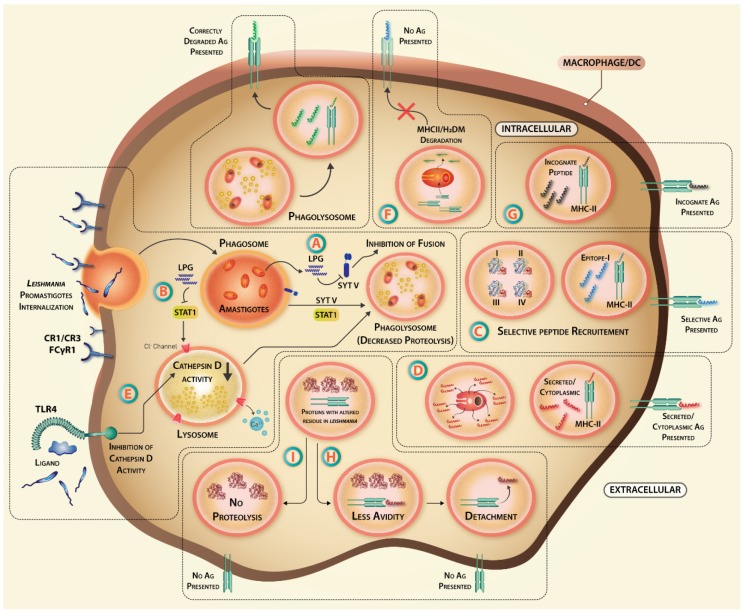
Leishmania-dependent factors affecting antigen priming leading to inefficacy of vaccination. Upon entry into the macrophages, (**A**) Leishmanial lipophosphoglycan (LPG) inhibits fusion of phagosome with lysosome via inhibition of SytV recruitment. (**B**) It also inhibits STAT-1, which is responsible for maintenance of conductivity of Cl- ion channels for phagosomal acidification following phagolysosome formation. (**C**) Amastigote-expressing proteins, which have high turnover during infection, can trigger intracellular competition among the peptides and influence determinant selection. (**D**) Cytoplasmic and secretory proteins of parasites have better chances of being processed and presented, inducing a bias towards the antigens to be predominantly presented to T cells. (**E**) TLR4-induced inhibition of Cathepsin D activity may reduce protein processing ability. (**F**) MHC-II and H2-DM molecules are taken up by amastigotes inside phagolysosome and degraded. (**G**) Infection-induced loading of incognate peptides instead of Leishmanial peptides can affect antigenic loading on MHC molecule. (**H**,**I**) Leishmanial antigens that may have substitutions in anchor residues or protease cleavage sites may either bind weakly to MHC-II, become resistant to action of proteases, or may function as altered peptide ligands, if selected for presentation to T cells.

**Figure 4 vaccines-07-00156-f004:**
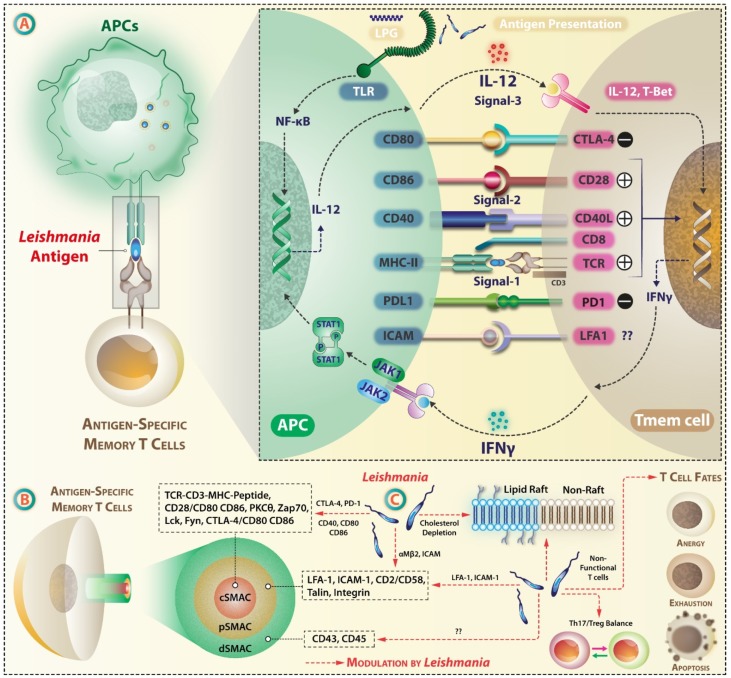
Represents formation of immune synapse (IS) between Leishmania infected APC and T cells and selective modulation of IS components by Leishmania. (**A**) On recognizing an APC with MHC bound cognate antigen, T cells form a contact zone called immune synapse (IS). IS comprises of the core supramolecular activation cluster –cSMAC, the peripheral contact zone-pSMAC, and the distal contact zone, each of which is characterized by the presence of unique cluster of differentiation (CD) molecules, and signaling intermediates like Zap70, each having distinct physiological roles in IS. (**B**) T cell activation occurs via reception of three signals: (1) Through the triad of TCR-MHC-Peptide-CD4, (2) via costimulatory molecules like CD28, and (3) by the action of cytokines. (**C**) Leishmania alters the membrane lipid content of infected APCs and may affect the clustering of signaling molecules. It targets costimulatory (CD28, CD80, CD86) molecules and coinhibitory molecules (PD-1, CTLA-4). By selective activation or deactivation of the pathways associated with these molecules, Leishmania can induce anergy, exhaustion, and apoptosis in target lymphocytes.

**Table 1 vaccines-07-00156-t001:** A comprehensive compilation of the vaccine approaches and analyses tested against *Leishmania*.

VACCINE CANDIDATE	STRAIN (CHALLENGE)	ROI	ADJUVANT	OUTCOME	DRAWBACK(S)	REF
**1ST GENERATION VACCINES**						
**Leishmanization (LZ)**	*L. major*	S.C	--	Higher success rate in Uzbekistan & Iran	Loss of infectivity, safety & ethical issues	[17]
**Killed but metabolically active**	*L. major* & *L. infantum chagasi*	S.C	--	Treating them with amotosalen and low dose of UV radiation	Shown promise in murine model but failed in human trials	[18]
**Killed but metabolically active**	*L. major* (Alum ppt. Autoclaved *L. major*)	--	BCG	Shown promise in VL and PKDL patients	Chances of relapse	[19]
**2ND GENERATION VACCINES**						
**FML (Fucose Mannose Ligand)**	*L. donovani*	S.C, I.P	Saponin, Al(OH)_3_, QuilA IL-12, BCG	Protective efficacy in BALB/C, Swiss albino, Hamster & dogs	Not shown promise in human	[20,21]
**Leishmune**	*L. chagasi*	S.C	--	No parasites were detected in blood, Skin and Lymph node	No trials in humans	[22]
**Leishmune^®^**	*L. donovani* *L. chagasi*	S.C	Saponin	Have transmission blocking property	Partial protection	[23]
**LAG (*L. donovani* promastigote membrane antigens)**	*L. donovani*	I.P	Liposome	Shown promise in mice and hamster	Failed in higher animal models	[24]
**SLA (Soluble Leishmanial Antigen)**	*L. donovani*	I.P, S.C	Liposome, MPL-TDM, non-coding pDNA+ISS	Prophylactic and therapeutic effect in mice model	Free or negatively charged liposome confers partial protection	[25,26,27]
**gp36**	*L. donovani*	S.C	Saponin	Protective effect in murine model	Not reached human trials	[28]
**Ld91, Ld72, Ld51, Ld31**	*L. donovani*	I.P	Liposome	Reduce parasite burden in visceral organ	Protective efficacy only in mice models	[29]
**dp72 and gp70-2**	*L. donovani*	I.P	*Corynebacterium parvum*	Decrease in parasitemia in visceral organ	Gp70-2 were not promising in mice	[30]
**SA (Soluble Antigen)**	*L. donovani*	I.M	CpG-ODN	Decrease in parasite load	Ag alone show mixed response in mice models	[31]
**Ribosomal P0 protein**	*L. major*	S.C	CpG-ODN	Shown promise in C57BL/6 mice	Failure in BALB/C mice	[32]
**F2 subfraction (97.4-68 kda)**	*L. donovani*	I.D	BCG	Th1 mediated cellular response in cured VL patients and hamsters	Not tested in clinical studies	[33]
**F2 sub-fraction (89.9-97.1 kDa)**	*L. donovani*	I.D	BCG	99% parasite inhibition were observed in hamsters	Specific characterization of antigen is needed for higher trials	[34,35]
**LPG and *Phlebotomus duboscqi* salivary gland lysates (SGLs)**	*L. major*	S.C	--	LPG alone provide protection in BALB/C mice	LPG+SGLs failed to provide protection	[36]
**LPG**	*Phlebotomus duboscqi* sandflies	--	--	Block the transmission of *Leishmania*	--	[37]
**gp63 (leishmanolysin)**	*L. donovani*	I.P	Cationic (DSPC) liposomes	Protective effect in BALB/C mice	Mixed responses	[38]
**Recombinant.gp63**	*L. major*	--	--	Recognition of fusion protein	Fusion protein did not protect the mice	[39]
**gp63**	*L. major*	I.V, S.C	--	BCG expressing gp63 only provide Protection	Limited up to mice only	[40]
**gp63**	*L. major*	S.C	CpG ODN + Liposome	Th1 type response	Adjuvant is needed for protection	[41]
**gp63**	*L. donovani*	I.M, S.C	CpG ODN	Evoke cellular and humoral response	No trials on higher animal	[42]
**H2B (Histone protein)**	*L. major*	S.C	CpG	Th1 type response	Only amino-terminal region confer protection	[43]
**H2B**	*L. major*, *L. infantum* patients	--	--	Whole H2B protein induces Th1 profile in individuals	Not reached clinical trials	[44]
**rORFF + BT1**	*L. donovani*	S.C	CFA, ICFA	Detection of high levels of anti-ORFF and anti-BT1 antibodies	Partial protection	[45]
**rORFF**	*L. donovani*	I.M	CpG ODN	Combination of CpG ODN with rORFF induces reduction in parasite load	Partial protection	[46]
**ORFF**	*L. donovani* (antimony sensitive & resistant)	I.M	--	UBQ-ORFF induces cellular & humoral response	Ubiquitin is required for protective effect	[47]
**Elongation factor 2 (LelF-2)**	*L. donovani*	I.D	--	Shown promise in Hamster	Not reach clinical trials	[48]
**rF14**	*L. donovani*	I.M	MPL	Reduce in parasite burden in spleen and liver	Partial protection	[49]
**HASPB1**	*L. donovani*	S.C	IL-12	Induces DC to produce IL-12	rHASPB1 immunization induces Th2 response	[50]
**78kDa**	*L. donovani*	S.C	MPL-A, Liposome rIL-12, ALD, FCA	Enhanced DTH response, reduction in infection rate of peritoneal macrophages	Antigen alone induces IgG1 isotypes	[51]
**LCR1**	*L. chagasi*	I.P, S.C	--	Protective effect in BALB/C mice	BCG-LCR1 I.P administration did not show protection	[52]
**Meta1 antigen**	*L. major*	I.M, S.C	IFA	MCP-3/meta 1 fusion partially shift towards Th1 type	Rec. protein Meta1 antigen induces Th2 response	[53]
**LirCyP1**	*L. infantum*	I.P	IL-12	Promote differentiation of helper and memory T cell	Partial protection	[54]
**Maxadilan (MAX)**	*L. major*	S.C, I.P	CFA IFA	Vaccination against Max elicits Th1 type response	MAX enhance blood flow and inhibit the immune response of the host	[55]
**LJM19**	*L. infantum chagasi*	I.D	--	Induces low parasite load and high IFN-γ/TGF-β ratio	Not reached clinical trials	[56]
**A2 protein**	*L. donovani*	I.P	*Propianibacterium* acnes	Production of IFN-ϒ response	Mixed Responses	[57]
**rLdγGCS**	*L. donovani*	S.C	NIV	Induces humoral response	Mixed Responses	[58]
**FUSION, HYBRID AND POLYPROTEIN VACCINES**						
**Chimeric Q protein**	*L. infantum*	S.C	--	Clearance of parasites from majority of the dogs	Studies during natural infection condition are required	[59]
**Amastigote cysteine proteases**	*L. major*	S.C	--	Hybrid CPA/B elicits protective response	Partial protection	[60]
**TAT-antigen fusion protein**	*L. major*	I.D	CpG ODN 1826	CD8^+^ T cells rapidly proliferate, activate *Leishmania*-specific Tc1 cells	Protective effect is observed in C57BL/6 mice	[61]
**rLeish-111f**	*L. major*	S.C	rIL-12 MPL-SE	Protective efficacy in BALB/C mice	Mixed IgG2a/IgG1 Ab response	[62]
**Leish-111f**	*L. infantum*	S.C	MPL-SE	99.6% reductions in parasite loads, completed phase 1 and 2 safety human trial	--	[63]
**LEISH-F1**	*L. donovani*	S.C	MPL-SE	Vaccine is safe, immunogenic in healthy subjects	Further trials are needed	[64]
**LIVE, ATTENUATED (GENETICALLY) VACCINES**						
**dhfr-ts^−^ null mutant**	*L. major*	I.V, S.C, I.M	--	Unable to cause disease in susceptible and immune deficient (nu/nu) BALB/C mice	Need to be tested on animal higher models	[65]
**dhfr-ts^−^*L. major* knock out mutant**	*L. amazonensis*	S.C, I.V	--	Cross- Protection was observed in both BALB/C and C57BL/6 mice	Partial degree of protection	[66]
**LdCen1(−/−)** ***L. donovani*** **centrin null mutants**	*L. donovani* *L. braziliensis*	I.M I.C	--	No parasites were seen in spleen, liver in BALB/C, SCID mice and Hamster, Cross protection against *L. braziliensis*	Not reach clinical trials	[67]
**(BT1−/−)** **Biopterin transporter null mutant**	*L. donovani*	I.V	--	Th1 mediated response were observed in BALB/C mice	Need to be tested on higher animal models	[68]
**LiSIR2(+/−)** ***L. infantum* SIR2 single knockout**	*L. infantum*	I.P	--	Reversion of T cell anergy, increased IFN-γ/IL-10 ratio	Mixed response in BALB/C mice	[69]
**Ldp27(−/−)**	*L. donovani* *L. infantum chagasi*	I.C	--	less COX (Cytochrome C oxidase complex) activity and ATP synthesis	Safety efficacy need to be tested on higher animal models	[70]
**ΔHSP70-II**	*L. major*	I.P, I.V, S.C	--	Promise protection in BALB/C model, consider safe in SCID mice and hamster	Need to be tested on higher animal models	[71]
**LiΔHSP70-II null mutant**	*L. major*	I.V, S.C	--	Promise protection in BALB/C and C57BL/6 mice	Limited only to mice models	[72]
**Δlpg2**	*L. major*	S.C	CPG-ODN	Shown promise in C57BL/6 mice	Antigen alone is not sufficient to provide protection	[73]
**lpg2^−^**	*L. major*	Foot pad	--	Suppress IL-4 and IL-10 responses	Very low IFN-ϒ response were observed	[74]
***Leishmania tarentolae*** **(Non-pathogenic)**	*L. donovani*	I.P	--	Activates DC, Induces IFN-ϒ, Th1 phenotype in BALB/C mice	Further testing is needed	[75]
**A2-recombinant** ***L. tarentolae***	*L. infantum*	I.P, I.V	--	Th1 response in BALB/C mice	I.V route elicits Th2 response	[76]
**A2-expressing *Lactococcus lactis***	*L. donovani*	S.C	--	Antigen-specific humoral response was observed	Only in BALB/C mice	[77]
**3RD GENERATION VACCINES**						
**LACK**	*L. major*	S.C	+rIL-12 -rIL-12	Protective response in BALB/C mice	Need to be tested on higher animal models	[78]
**pCIneo-LACK**	*L. chagasi*	I.N Intranasal	--	Reduce parasite burden in liver and spleen	Limited up to BALB/C mice	[79]
**LACK-DNA**	*L. chagasi*	I.N	--	Cellular and humoral response were observed	Not reached clinical trials	[80]
**p36 (LACK) DNA**	*L. donovani*	I.D, S.C	IL-12	Robust Th1 response, highly immunogenic	No protection	[81]
**gp63 and Hsp70**	*L. donovani*	S.C	--	Significantly reduce the parasite burden	Need to be tested on higher animal models	[82]
**A2+E6**	*L. donovani*	I.M	--	Induce cellular and humoral response, inhibit cellular p53 response	Limited to mice model	[83]
**A2 (Expressed in adenovirus)**	*L. chagasi*	S.C	--	Reduced parasitism in spleen and liver, production of high level of IFN-ϒ	Low antibody response	[84]
**ORFF**	*L. donovani*	I.M	--	Induces both cellular and humoral response	Limited to BALB/C mice	[85]
**rORFF**	*L. donovani*	I.M	IL-12	Higher levels of IFN-γ were observed	Need to be tested on higher animal models	[86]
**KMP-11**	*L. donovani*	I.M	--	Reversal of T-cell anergy with functional IL-2 generation	Mixed response	[87]
**KMP-11**	*L. major*	I.M, S.C	IL-12	Robust IFN-ϒ production, polarized Th1 response	Require adjuvant for complete protection	[88]
**ϒ GCS**	*L. donovani*	I.M	--	Elevated humoral and cell mediated response	Limited to mice model only	[89]
**H2A, H2B, H3, H4, LACK**	*L. donovani*	I.M, I.D	IL-12 GM-CSF	Induce DTH response, reduction in parasite burden	Unable to induce parasite-specific antibody response	[90]
**N-terminal domain of proteophosphoglycan (PPG)**	*L. donovani*	I.M	--	Induce IFN-ϒ, TNF-α, IL-12 and downregulate TGF-β, IL-4, IL-10	Need further trials ahead of hamster model	[91]
**Single antigen Gp63, polytope and polytope HSP70**	*L. donovani*	I.M, I.P	--	Strong Th1 response	Limited to mice only	[92]
**NH36**	*L. chagasi* *L. mexicana*	I.M, S.C	--	Reduction in parasite load, Th1 mediated response, cross-protection	No differences were observed after infection with *L. mexicana*	[93]
**NH36**	*L. chagasi*	I.M, I.P	--	Reduction in parasite burden in liver	Clinical trials not done	[94]
**KMP11, TSA, elongation factor P74, CPA and CPB, HASPB, A2 LEISHDNAVAX**	*L. donovani*	I.D	--	Induce CD4^+^ and CD8^+^ T cell responses in genetically diverse human populations	Require clinical trials	[95]
**KMPII, TRYP, LACK and GP63**	*L. infantum*	I.D	--	Vaccine was safe and well tolerated for dogs	No protection in dogs	[96]
**papLe22**	*L. infantum*	I.M	--	Downregulate Th2 type response in hamsters	Low antibody response	[97]
**HbR**	*L. donovani*	I.M	--	Complete protection in both BALB/C and Hamsters	Clinical trials not done	[98]
**Tuzin**	*L. donovani*	I.M	--	higher levels of IFN-γ and IL-12 production	Limited only to BALB/C mice	[99]
**HETEROLOGOUS PRIME-BOOST (HPB) VACCINES**						
**ORFF**	*L. donovani*	I.M	Alum	75%–80% reduction in parasite load, enhanced production of IgG2a and IFN-γ in BALB/C mice	Need to be tested on higher animal models	[100]
**Cysteine proteinases type I and II**	*L. infantum*	S.C	CpG ODN and Montanide 720	Antigen-specific immune response was observed	Need to be tested on higher animal model	[101]
**rLdP1**	*L. donovani*	I.M	FCA	75.68% decrease in splenic parasite burden in hamsters,	Challenge by other *Leishmania* species needed for evaluation	[102]
**A2-CPA-CPB (CTE) recombinant *L. tarentolae***	*L. infantum*	Foot pad	cationic solid lipid nanoparticle (cSLN)	High NO production, and low parasite burden	Mixed responses	[103]
**LACK-WR or LACK-MVA**	*L. infantum*	I.D, I.P	--	High levels of protection in the draining lymph node	Low protection observed in Spleen and Liver	[104]
**AlphaGalCer + DNAp36+ VVp36**	*L. major*	I.D, I.P	αGalCer	20-fold reductions in parasite burdens	Limited to mice only	[105]
**DNA/MVA TRYP**	*L. Viannia panamensis*	I.D	α-GalCer, LPS, CpG, Pam3CSK4, MALP-2	TLR1/2 activation, antigen specific CD8 cells	Need to be tested on higher animal model	[106]
**pORT-LACK/MVA-LACK**	*L. infantum*	S.C	--	increase in expression of Th1 type cytokines in PBMC and target organs	Limited to canine model only	[107]
**LdPxn-1 (*L. donovani* Peroxidoxin-1)**	*L. major*	I.M S.C	m-GMCSF	Induction of multipotent CD4^+^ cells	Mixed responses	[108]
**DENDRITIC CELL(DC)-BASED VACCINES**						
**DCs pulsed with peptide L1 of gp63 (SPL) & pulsing with peptide L2 of gp63**	*L. major*	I.V	--	Reduce lesion and parasite load, Th1 phenotype in BALB/C mice with L1 peptide	Partial protection with L1 peptide, whereas L2 peptide switch towards Th2 profile	[109]
**Plasmacytoid DCs**	*L. major*	I.V	CpG-ODN	Induce T cell-mediated protection	IFN-α production could not be detected	[110]
**SLDA-pulsed DCs**	*L. donovani*	I.V	IL-12	Live parasites were not detected in the liver of mice, 1-3 log lower parasite burdens	Granuloma formation were found in SLDA-pulsed, non-transduced DCs BALB/C mice	[111]
**BM-DCs pulsed with KMP-11 (12–31aa) peptide**	*L. infantum*	I.V	CpG-ODN	Induced the differentiation of peptide-specific Th17 cells	Mixed responses	[112]
**SLDA-pulsed syngeneic BM-DCs**	*L. donovani*	I.V, I.M	--	Complete clearance of parasites from spleen and liver	Need to be tested on higher animal models	[113]
**Hybrid cell vaccine (HCV)**	*L. donovani*	I.V	--	Strong antigen-specific CTL generation	Expression of IL-4 and IL-13 get elevated at both transcription and translational levels	[114]

ABBREVIATIONS: ALD (Autoclaved Leishmania antigen), BCG (Bacille Calmette Guerin), CFA (Complete Freund’s Adjuvant), CpG-ODN (CpG- oligodeoxynucleotides), DSPC (Distearoyl phosphatidylcholine), FCA (Freund’s adjuvant), I.D (Intradermal), I.M (Intramuscular), I.N (Intranasal), I.P (Intraperitoneal), I.V (Intravenous), ICFA (Incomplete Freund’s adjuvant), IFA (Incomplete Freund’s Adjuvant), ISS (Immunostimulatory sequences), LPS (Lipopolysaccharide), m-GMCSF (murine granulocyte-macrophage colony-stimulating factor), MPL-SE (Monophosphoryl lipid A plus squalene), MPL-TDM (Monophosphoryl lipid-trehalose dicorynomycolate), NIV (non-ionic surfactant vesicle), S.C (Subcutaneous).

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
