# Peer review of "Anti-Leishmanial Vaccines: Assumptions, Approaches, and Annulments"

_vaccines, 2019, doi:10.3390/vaccines7040156_

Round 1

Reviewer 1 Report

The manuscript Anti-leishmanial vaccines: assumptions, approaches and annulments presented a good revision regard the tentatives of Leishmania vaccines development. 

The strength points of the manuscript are the organization and the exhaustively cover of the major effort that researchers are made to try a leishmania vaccine. The weakness that I observed is a lack of precision in the representation of parasite forms promastigote/amastigote in one figure as well the connection of T cells with leishmania dead.

In my opinion, the manuscript can be accepted after minor revision as follows:
1) Figure 1 must be improved because have a misunderstanding that T cells kill Leishmania: TCD4 cell activated macrophage to kill the parasite while TCD8+ kill the infected cells. Please, change the figure to explain better the role of T cells in Leishmania vaccine.
2) The Leishmania in figure 1 figure seems to be promastigotes instead of amastigotes. The authors must change the representation of parasite promastigotes to amastigotes.

Reviewer 2 Report

Authors review several published aspects associated to lack of response to vaccination in leishmania. A comprehensive review of vaccination approaches and outcomes, in animal models and endemic area, was shown in the main Table.

Authors describe hypothetical parameters responsible for failure to leishmania vaccination. Those parameters were extensively reviewed and included in figures and text.

In addition to known mechanisms of vaccine failure, authors explored less studied mechanisms, that may impact leishmania vaccination, such as effect of inhibitory receptors in immunological synapse. As far my knowledge assertions in text and figures were supported by references.

Minor modifications:

Line 42 reference to immune parameters responsible for vaccine failure is schematized in Figure 3 (not Figure 1).

Figure 3. Improve resolution of Figure. Connect decreased cathepsin with decreased lysosomal proteolysis and decreased MHCII peptide binding.

Figure 4. C, missing. Separation of pictures B, C and legend could be improved.

Reviewer 3 Report

The authors present an authoritative overview of the challenges that we face in developing an effective vaccine against Leishmania. The authors have undertaken an extensive review of the Leishmania vaccine literature and presented it well with the summary table being especially useful. Moreover, the cartoons are excellent and act as informative summaries of the text.

As a summary of the literature this is a really good review; however, my major criticism is that the conclusion at the end of the manuscript was disappointing as the authors did not take the opportunity to clearly highlight what they believed to be the most promising approach and what is in their opinion the biggest stumbling blocks in the way of vaccine production.

Overall the quality of the English was good but in places they were lapses and I’ve tried to identify them all but I would recommend a native English speaker double check the revised manuscript.

Specific comments

Line 3 – annulments is not appropriate in this context – I understand the authors wanted another word beginning with ‘a’ but it can annulments and it needs to be revised.

Line 10-12 – here the authors have an out of place 2 sentences starting ‘Leishmanization… ending …host-protective’ I would recommend deleting these sentences as they don’t fit with the rest of the abstract or revising substantially.

Line 21/22 – should this read ‘not cross-reactive’? as I would have thought this would lead to suppression.

Line 40 – delete ‘analyzing the failures in anti-leishmanial vaccines’

Line 40/41 – would read better if – ‘Here, we follow the scheme of immune priming, reactivation, and outcome of challenge infection…’

Line 49-51 – the end of this sentence is confusing and needs to be rephrased.

Line 56 – should read ‘...prompted the inclusion of the genes...’

Line 57 – there needs to be more explanation of the polarization of Th1/Th2 responses – i.e. is one better for protection than the other, should a vaccine try and stimulate one rather than the other.

In the table what is a DC-based vaccine – please spell out?

Plus, how are the vaccine generations defined? What makes something a primary, secondary generation vaccine?

Line 72 – which cell type/species are you referring to?

Line 74 – significant not significantly

Line 76 – 'their preferred' not 'the preferred'

Line 127/128 – would read better if, ‘…amastigote form, inhibiting the antigen presentation process and affecting…’

Line 182/183 – would read better if, ‘…THP1 macrophages with the infected macrophages having different HLA-I peptides [142] implying…’

Line 187 – hindered not hindering

Line 215 – delete ‘via regulation of gene expression’

Line 244 – would read better if ‘…or by the action of certain extracellular…’

Line 272 – delete ‘gene’

Line 284 – Leishmania not Leishmanial

Line 289 – delete ‘for impairment of IS’

Line 301/304 – the final 2 sentences of this section don’t fit well. In the first sentence you refer to 3 signals but these are not clearly highlighted above. This sentence needs to be rephrased so it is integrated with work above it.

The second sentence about T cell fates feels as if it should be integrated into the next section or the one after and those sections feel complete so this sentence can easily be deleted without loss of meaning.

Line 339 – cytokine not cytokines

Line 398/390 – this figure 4 reference feels more appropriate for the immune synapse section. It can be referenced here as well but should be first mentioned in the IS section.

Line 404 – the conclusion section is weak as I have previously mentioned.

The authors talk about T cell migration but this feels more like something that should have been dealt with in the main body of the text or is this one of a series of an issues there isn’t space to deal with?

There’s a secondly etc without a firstly.

The points raised after secondly, thirdly etc seem to point to areas where more work is needed but the authors need to be more explicit about this to really drive the point home.
